



# The limited effect of deforestation on stabilized subsoil organic carbon in a subtropical catchment

Claude Raoul Müller[1], Johan Six[1], Liesa Brosens[2,3], Philipp Baumann[1], Jean Paolo Gomes Minella[4],

Gerard Govers[2], Marijn Van de Broek[1]

[1]Department of Environmental Systems Science, ETH Zurich, 8092 Zürich, Switzerland
[2]Department of Earth and Environmental Sciences, KU Leuven, 3000 Leuven, Belgium
[3] Environmental Modeling Unit, Flemish Institute for Technological Research (VITO), 2400 Mol, Belgium
[4]Department of Soil, Universidade Federal de Santa Maria, 97105-900 Rio Grande do Sul, Brazil

*Correspondence to*: Claude R. Müller (claude.mueller@usys.ethz.ch)

**Abstract.** Predicting the quantity of soil organic carbon (SOC) requires understanding about how different factors control the amount of SOC. Land use has a major influence on the function of the soil as a carbon sink, as shown by substantial organic carbon (OC) losses from the soil upon deforestation. Yet, predicting the degree to which land use change affects the SOC

content, and the depth down to which this occurs, requires context-specific information related to, for example, climate, geochemistry, and land use history. In this study, soil samples collected down to 300 cm depth from forests and agricultural fields in a subtropical region (Arvorhezina, southern Brazil) were used to study the impact of land use on the amount of stabilized OC along the soil profile. We found that the stabilized SOC content was not affected by land use below a depth of 90 cm. Along the soil profile, the amount of stabilized OC was predominantly controlled by land use and depth, in addition to

the silt and clay content, and aluminum ion concentrations. Below 100 cm, none of the soil profiles reached a concentration of stabilized SOC above 50% of stabilized SOC saturation point (i.e., the maximum OC concentration that can physically be stabilized in these soils). Based on these results, we argue that it is unlikely that deeper soil layers can serve as an OC sink over a time scale relevant to global climate change, due to limited OC input in these depth layers. Furthermore, we found that soil weathering degree was not a relevant control on the amount of stabilized SOC in the profiles we investigated, because of

the high weathering degree of the studied soils. It is therefore suggested that while the soil weathering degree might be an effective controlling factor of OC stabilization over large spatial scale, it is not an informative measure for this process at the scale of the soil profile in highly weathered soils.



## 1   Introduction


Soils contain an estimated amount of 1.500 gigatons of organic carbon (OC) in the upper meter (excluding permafrost) (Balesdent et al., 2018; Jobbágy and Jackson, 2000; Scharlemann et al., 2014). It has been estimated that ca. 44 % of global SOC is stored in the tropics (Veldkamp et al., 2020), which are undergoing the greatest rate of land-use change globally, mainly as a consequence of the rapid conversion of natural land uses to agricultural land (Hansen et al., 2013). This rapid land use

change has considerable impacts on the amount of soil organic carbon (SOC) stored in tropical ecosystems. Indeed, the conversion of tropical forests to agricultural fields leads to an average decrease in the OC concentration in the topsoil (< 30 cm depth) of up to 50% after 25 years (Veldkamp et al., 2020).

Nevertheless, estimates of changes in OC vary a lot between studies (de Blécourt et al., 2019; Bonini et al., 2018; Comte et

al., 2012; Detwiler, 1986; Don et al., 2011; Powers et al., 2011) because this depends on different factors, such as the time since land use conversion and agricultural management practices (e.g., residue inputs reduce SOC loss after conversion) (Detwiler, 1986; Veldkamp et al., 2020). While it is commonly accepted that land use has an impact on the amount of SOC, it is unclear down to which depth SOC storage is affected by land use change for specific climatic regions. Therefore, estimates of the effect of LU change on the SOC stock below a depth of 30 cm are uncertain. Although some studies only observed a

change in the topsoil (e.g., Guillaume et al., 2015), Strey et al. (2016) argue that soils should be studied at least down to 100 cm depth to fully assess the impact of land use change on the SOC stock. For example, Veldkamp et al. (2020) found that the forest soils below 0.5 m depth contained on average 35% more SOC compared to crop plantations that had been converted for more than 25 years. This illustrates that the subsoil needs to be accounted for when assessing the impact of land use change on the SOC content.


The soil organic carbon pool consists of a wide variety of organic molecules, ranging from unprocessed plant material to organic molecules with a wide range of oxidation states (Kögel-Knabner, 2002; Lehmann and Kleber, 2015). The residence time of OC in soils is not uniform, with respired $CO_2$ generally being substantially younger than bulk SOC (Trumbore et al., 1995), which can have an age in the order of thousands of years (Mathieu et al., 2015). As unprocessed organic residues

decompose, they break down into smaller OC-containing particles that can form chemical bonds with soil minerals, a process that at least partially protects OC from microbial degradation and thus contributes to long-term carbon storage (Lehmann, J. & Kleber, 2015; Cotrufo et al., 2013; Kögel-Knaber et al, 2002; Schrumpf et al., 2013; von Lützow et al., 2008). It is estimated that this mineral-associated OC (MAOC) constitutes more than 65% of total SOC globally (Georgiou et al., 2022).



To quantify the proportion of SOC pools with different properties and a different turnover rate, soil organic matter (SOM) is generally separated into particulate organic matter (POM), which consists of relatively unprocessed plant material, and mineral-associated organic matter (MAOM) (Cotrufo et al., 2019; Elliot et al., 1992; Lehmann and Kleber, 2015; Kögel-Knabner et al., 2008) using a variety of soil fractionation methods (Poeplau et al., 2018). POM is characterized by a low density, a high C:N ratio, and a relatively young age, while MAOM has a high density, a low C:N ratio, and a relatively old

age (Kögel-Knabner et al., 2008; Lavallee et al., 2020; von Lützow et al., 2008). Information on how SOC is distributed among these pools is useful, for example, to assess how agricultural management practices can store atmospheric $CO_2$ in SOC pools with a long residence time (Chenu et al., 2019; Kallenbach et al., 2015; Kögel-Knabner et al., 2022; Tiefenbacher et al., 2021), or to inform SOC models that simulate the proportion of SOC in pools with different turnover rates (Abramoff et al., 2018; Ahrens et al., 2015; Tang and Riley, 2015).


The dynamics of MAOC have mainly been studied in the topsoil (here defined as the upper 30 cm of the soil) in temperate regions (Cotrufo et al., 2019; Georgiou et al., 2022; Lugato et al., 2021; Kleber et al., 2015; Kramer and Chadwick, 2018; Rocci et al., 2021; Sokol et al., 2022). At the scale of the European continent, for example, it has been shown that between ca. 25 and 100 % of topsoil SOC consists of MAOC, depending on land use and type of mycorrhizal association of the dominant

vegetation(Cotrufo et al., 2019; Lugato et al., 2021). However, the subsoil (≥ 0.30 m depth) contains a large amount of the total SOC stock (ca. 55 % of total SOC stock globally (Lal, 2018)). This OC has average residence times ranging from decades to millennia (Mathieu et al., 2015), making it an important long-term carbon sink in terrestrial ecosystems (Rumpel and Kögel-Knabner, 2011). Subsoil OC consists of more processed and microbially-derived molecules with a lower C/N ratio compared to the topsoil (Schrumpf et al., 2013). These OC molecules are small and abundant in polar groups (Kleber et al., 2021).

Thereby, they are highly reactive towards the mineral matrix, causing the share of MAOC to total organic carbon (TOC) to generally increase with increasing soil depth (Cotrufo et al., 2013; Schrumpf et al., 2013).

Improving global projections of SOC stabilization requires predictive models that allow to assess the amount of MAOC that currently is and potentially can be stored in soils under specific climate and land use regimes. Building such models requires

that the factors controlling MAOC are understood. For example, soils with high silt (soil particles < 53 μm and > 2 μm) and clay (soil particles < 2 μm) contents generally store more SOC because of their higher specific surface area as compared to coarse-textured soils (Amato and Lass, 1992; Feller and Beare, 1997; Hassink, 1994; Kleber et al., 2015). However, there is an upper limit to the capacity of soil minerals to stabilize organic molecules (Hassink, 1997; Six et al., 2002c), generally referred to as the maximum sorption capacity (Guggenberger and Kaiser, 2003; Kothawala et al., 2009). The latter has been

shown to be controlled by edaphic factors such as mineralogy and grain size (Abramoff et al., 2021; Feng et al., 2013; Rasmussen et al., 2018). Across the soil profile, the ratio of MAOC to maximum sorption capacity (i.e., the MAOC saturation) generally decreases with soil depth, with subsoils having the largest MAOC deficit, and thus, at least theoretically, the largest potential to stabilize additional OC (Chen et al., 2018; Georgiou et al., 2022). Therefore, it has been proposed that increasing



the amount of subsoil MAOC can lead to a net transfer of atmospheric $CO_2$ to the SOC pool for timescales of decades to
millennia (Rumpel et al., 2020), e.g. through growing deep-rooting crops (Kell, 2012), deep tillage (Alcántara et al., 2016), or
deep soil flipping (Schiedung et al., 2019). While the potential for additional OC storage in the subsoil has been relatively well
studied for different soil types and ecosystems in temperate ecosystems, data for tropical ecosystems is scarce (Georgiou et
al., 2022).

In addition to soil texture, mineralogical and morphological characteristics of the soil have been identified as controls on SOC
content (Rasmussen et al., 2018), while polyvalent cations such as $Al^{3+}$ enable negatively charged OC molecules to bind to the
mineral surface (i.e., cation bridging)  (von Lützow et al., 2006). Furthermore, mineral soil weathering, which is the degree of
degradation of bedrock and minerals due to physical, chemical, and biological factors, can have either a positive or a negative
control on SOC stabilization. On the one hand, it promotes the formation of OC - mineral associations by increasing the specific
surface area (SSA) of soil particles or by releasing Al and Fe oxides and cations (Depetris et al., 2014; Kleber et al., 2015). On
the other hand, a high weathering degree might alter the structure of secondary minerals, reducing their reactivity and thus the
potential for OC stabilization (Doetterl et al., 2018). Yet, all these parameters might influence each other, and their interaction
depends on the climate and geochemistry (Doetterl et al., 2015a). For example, Six et al. (2002b) observed a lower association
between silt and clay particles and OC in tropical soils in comparison to temperate soils. Therefore, it is necessary to study
specific environments, such as highly weathered tropical soils, in order to disentangle the complex relationship between SOC
storage and abiotic factors. Without a proper understanding of the controls of SOC stabilization in tropical soils, the
extrapolation of the findings from a limited number of field observations may lead to erroneous results (Powers et al., 2011).

Here, we studied the effect of forest conversion to agriculture on stabilized OC along the soil profile in a subtropical
environment. We analysed how deforestation in a subtropical catchment in southern Brazil has affected OC associated with
the silt and clay fraction down to 3 m depth. The contents of OC associated with silt and clay within the bulk soil is referred
to as the silt and clay organic carbon ($OC_{S+C}$) and represents OC stabilized on soil minerals (i.e, MAOC). The main factors
controlling the amount of $OC_{S+C}$ along the soil profile were assessed using a general additive mixed effect model (GAMM).
The silt and clay contents were used as limiting parameters to estimate the stabilized OC saturation point (i.e., the maximum
OC concentration that can physically be stabilized in these soils). We hypothesized that $OC_{S+C}$ concentrations differ between
the two land use types in the topsoil, but that this difference is no longer present at below 100 cm depth. We furthermore
hypothesized that soil texture and weathering degree are important controlling factors of the $OC_{S+C}$ concentration, and that
$OC_{S+C}$ in the subsoils of both environments is well below the maximum sorption capacity of the soil.



## 2 Material and Methods

### 2.1 Study Design and Soil Sampling

The study area is located in southern Brazil at the southern edge of the Paraná Basin in the "Serra Gerral" region (state of Rio Grande do Sul) (Fig. 1). The climate is classified as warm and humid with no dry season (Köppen classification Cfb, (Alvares et al., 2013; Peel et al., 2007).), with precipitation being evenly distributed throughout the year with an annual mean of 1900 mm, while the mean monthly temperature varies from 12°C in July to 22°C in January (Robinet et al., 2018a, b). The elevation is between 370 and 790 m above sea level and the slopes vary between 2.5° and 30° (Brosens et al., 2021). The dominant soils in the study area are Acrisols, Leptosols, and Cambisols, overlying a basalt and dacite-rhyolite bedrock (Turner, Simon et al., 1994; Caner et al., 2014; Minella et al., 2009). The natural vegetation of the study area consists of a mixed Ombro-philous forest (Araucaria forest), that has been subject to intensive land use change from forest to agriculture since the beginning of the 20th century (Morellato and Haddad, 2022; Lopes, 2006). The cultivated crops on the agricultural fields where samples were collected were tobacco (Nicotina tabacum), erva-maté (Ilex paraguariensis), mais (Zea mays), and soybean (Glycine max).

In order to assess how deforestation affected the distribution of $OC_{S+C}$ down the soil profile, soil samples collected by Brosens et al. (2020, 2021) were used. These comprised a total of 226 soil samples collected on mid-slope positions in 46 forested locations and 48 agricultural fields using an Edelman auger. Sample collection is briefly described here, for more details we refer to the original studies. Sampling locations were chosen randomly in a circular region of 250 km$^2$ (Fig. 1), using an approach that aimed to sample the widest variation in hillslope gradients in the study area. Only agricultural soils that were converted from forest for at least 30 years were selected (Supplementary Table 3). At each sampling location, one soil sample was taken from each soil horizon down to the bedrock or saprolite, with a maximum sampling depth of 2.95 m. The number of samples collected per site depended on the number of soil horizons, which ranged between 1 and 6. A sample was collected from the center of the augered soil to avoid contamination from more shallow soil layers.



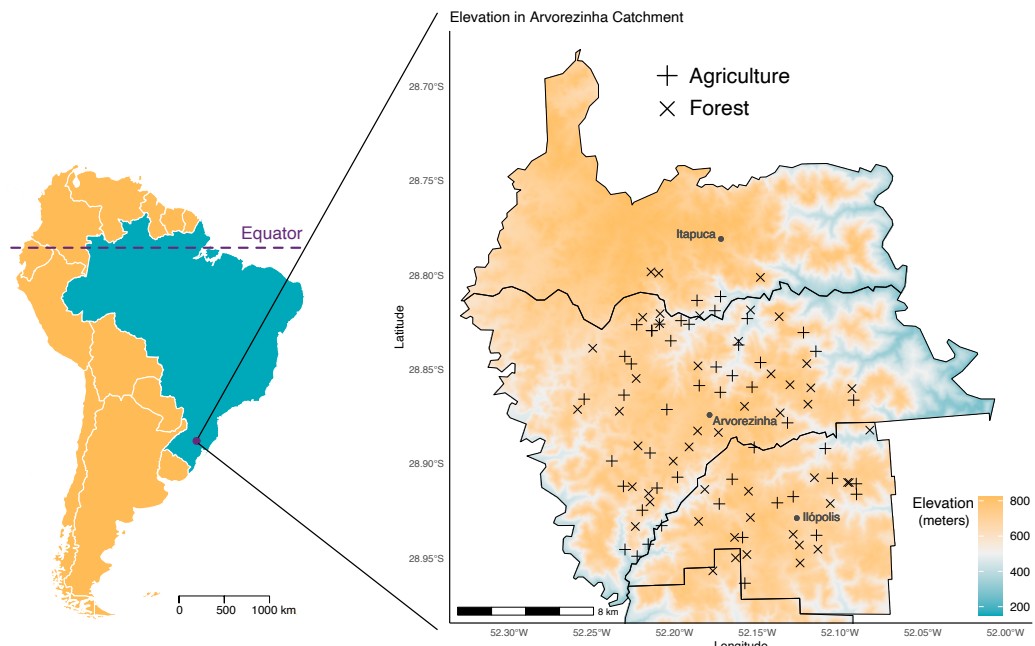

Figure 1: Map showing sampling location. Location of the study area in South America (left) and the municipalities of Arvorezinha, Itapuca, and Iliópolis (right) with indication of the sampling locations (46 under forest and 48 under arable land). The elevation map was obtained from the *R* package *elevatr (Hollister et al., 2022)*.

## 2.2 Measurements and laboratory analysis

### 2.2.1 Sample selection

All laboratory analyses were performed on a subset of samples from 40 selected sites (20 sites for each land use), with the exception of the mid-infrared spectra (see below), which were measured on all 229 samples from the 94 sites. The sites from which the samples were analysed in the laboratory (the 40 selected sites) were selected according to 3 criteria: location (evenly distributed throughout the study area), soil depth (representing all possible maximum soil depths evenly), and the weathering degree (all levels of weathering degree are represented evenly).

### 2.2.2 Measuring mid-infrared spectra

After drying, sieving, and grinding, the soil samples were measured with mid-infrared reflectance spectroscopy (MIR, FrontierTM with auto-sampler from PerkinElmer, Waltham, MA, USA) by Brosens et al. (2021). The measured wavenumbers ranged from between 400 and 4000 cm$^{-1}$, with a resolution was 1 cm$^{-1}$). Signal scattering was removed, and data quality and





calibration were improved using various pre-processing treatments. More details about these analysis can be found in Brosens
     et al. (2021).

### 2.2.3    Weathering indices

     Brosens et al. (2021) used mid-infrared spectroscopy to predict multiple weathering indices of the collected soil samples, using
the samples of Vanacker et al. (2019) that were collected in the same catchments to calibrate their spectral model. Two soil
     weathering indices measured by Brosens et al. (2021) are used in this study: the total reserve in bases (TRB, [cmol$_c$ kg$^{-1}$]) and
     the chemical index of alteration (CIA, dimensionless). The chemical index of alteration reflects the degree of weathering as
     the relative proportion of $Al_2O_3$ (i.e., a conservative oxide) to the sum of $Al_2O_3$, CaO, $Na_2O$, and $K_2O$ (i.e., major oxides),
     which can be interpreted as the extent of the conversion of feldspar to clay minerals by evaluating the mobility of the cations
$Ca^{2+}$, $K^+$ and $Na^{2+}$  (i.e., Eq. (1)) (Burke et al., 2007; Nesbitt and Young, 1982). The higher the CIA, the more labile cations
     were set free and the higher the weathering degree.

$$CIA = 100 * \left[\frac{Al_2O_3}{Al_2O_3 + CaO + Na_2O + K_2O}\right],$$
(1)

The total reserve in bases (TRB, [cmol$_c$ kg$^{-1}$]) is the sum of alkaline and alkaline-earth elements (i.e., $Ca^{2+}$, $Mg^+$, $K^{+,}$ and $Na^{2+}$)
     cations present in the soil (basic cations) (i.e., Eq. (2)). The lower the TRB, the higher the weathering degree (Delvaux et al.,
     1989).

$$TRB = Ca^2 + Na^+ + K^+ + Mg^{2+} \ [cmol_c * kg^{-1}],$$
(2)

### 2.2.4    Soil organic carbon fractionation

     To quantify the portion of SOC associated with soil minerals, OC fractionation was performed on the soil samples collected
     by Brosens et al. (2020). The OC associated with fine soil particles (< 53 μm, referred to here as silt and clay organic carbon,
     OC$_{S+C}$) was separated from the bulk soil (Del Galdo et al., 2003; Six et al., 2002a). The resulting OC$_{S+C}$ represents the OC
     protected by minerals in the soils (i.e., MAOC) (Cotrufo et al., 2019; Lavallee et al., 2020).
     To fractionate the soil samples, 10 g of dried and 2 mm sieved bulk soil was weighed in a 50 mL Falcon tube. Next, 30 mL of
     0.5% Sodium hexametaphosphate (NaHMP) and 20 glass beads were added, and the vial was shaken on a reciprocal at 150
     rpm for 18 hours. The dispersed sample was poured over a 53 μm sieve in a basin (30 cm diameter and 8 cm deep). The soil
     remaining on the sieve was thoroughly rinsed with de-ionized water until the water passing through the sieve was clear. The
soil remaining on the sieve (i.e., the soil fraction > 53 μm, referred to as the POM fraction) was poured into a pre-weighed



aluminium tray. The silt and clay soil fraction (i.e., the fraction < 53 μm, referred to as the S&C fraction) was transferred into a pre-weighed aluminium tray. Both soil fractions were dried at 90°C for 48h. Afterward, the aluminium trays were weighed to obtain the weight of both soil fractions.

### 2.2.5 Total and stabilized soil organic carbon

Prior to analysing the soil samples for OC% and total N, large organic particles were manually removed from the samples using tweezers. Afterward, 200 mg of dried and finely ground soil was analysed for OC% and total N using a CHN628 Series Elemental Determinator (LECO Corporation). The OC concentration of the non-fractionated bulk soil was also determined. As the pH was lower than 7 for all soil samples, it was assumed that the soil did not contain any carbonates. The carbon concentration of the S&C fraction was obtained from the fraction < 53 μm. The OC concentration of the S&C fraction in the

bulk soil is referred to here as silt and clay organic carbon ($OC_{S+C}$). It is defined as follows: the $OC_{s+c}$ is the mass of OC in the S&C fraction divided by the mass of bulk soil (Fig. S1). When the $OC_{S+C}$/TOC ratio was >1, it was assumed that 100% of the OC was $OC_{S+C}$. As bulk density was not measured on the samples, the OC stocks could not be calculated. Variations in the OC content of multiple samples are expressed as the standard deviation.

### 2.2.6 pH

To obtain $pH_{H2O}$, 10 g of dried soil was weighed in a Falcon tube, and 25 mL of deionized water was added. The vials were shaken on a reciprocal shaker at 150 rpm for 2 h. The slurry was allowed to settle for 24 h, after which the pH was measured using a pH meter (Thermo Scientific™ Eutech™ "150 Series Waterproof Handheld Meters").

### 2.2.7 Cation exchange capacity and aluminium cations

Before measuring the cation exchange capacity (CEC) and aluminium cations ($Al^{3+}$), visible organic particles were manually
removed from the soil sample using tweezers. Next, 2 g of dried and grinded soil was weighed in a centrifugation tube, 25 mL 0.1 M $BaCl_2$ was added and the samples were shaken on a reciprocal shaker at 150 rpm for 2 h. Next, the sample was centrifuged for 10 min at 2500 rpm and the supernatant was passed through a Whatman 41 filter. After filtering, 1 mL of the solution was diluted in 4 ml water, which was analysed with an ICP-OES (G8010A Agilent 5100 SVDV ICP-OES, Parent Asset SYS-10-5100). The concentration of Al, Ca, K, Mg, Mn, and Na cations in the solution was converted to $cmol_c$ $kg^{-1}$ soil

before calculating the effective cation exchange capacity by summing the above-mentioned cations (Hendershot & Duquette, 1986).

### 2.2.8 Soil texture

Prior to the analysis of soil texture, visible organic particles were manually removed using tweezers. Aggregates were gently crushed manually, and the soil was sieved to 2 mm. Next, between 200 and 300 mg of soil was put in 5 mL of 5 % $(NaPO_3)_6$
and shaken for 4 h, after which the solution was sonicated for 1 min. The soil grain size was determined using an LS13320



(Beckman coulter). Grain size classes were as follows: clay (< 2 μm), silt (> 2 μm and < 53 μm), and sand (> 53 μm and < 2 mm). To assess if the presence of organic matter in the samples affected the measured grain size, 20 samples were measured twice: (1) by treating them with 40 mL $H_2O_2$ before analysis and (2) without $H_2O_2$. As the results of both treatments showed no significant differences, the remaining samples were not treated with $H_2O_2$ prior to measurement, and dispersion with
$(NaPO_3)_6$ was the only pre-treatment.

## 2.3 Mid-infrared estimations of total and stabilized organic carbon concentrations

MIR spectra were used to determine the OC concentrations of the samples that were not analysed with the dry combustion method. To that end, a spectral model of the concentration of total organic carbon (i.e., TOC) and stabilized organic carbon (i.e., $OC_{S+C}$) was constructed using the MIR spectra of selected samples (n = 116). This model was used to predict the TOC
and $OC_{S+C}$ concentration of the remaining samples (n = 113). The spectral processing and modeling for both TOC and $OC_{S+C}$ was done with the R package simplerspec (Baumann, 2019). The spectra were averaged for 3 replicate measurements for each soil sample. Then, a partial least square regression (PLSR) calibration model was developed. For the final model, the absorbance spectra were pre-processed using a Savitzky-Golay first derivative with a window of 21 points (Savitzky and Golay, 1964). The models were tuned and evaluated using 10-fold cross-validation. The resulting models for TOC and $OC_{S+C}$
had an $R^2$ of respectively 0.94 and 0.88, and an RMSE of 0.22 (% OC) and 0.26 (% of $OC_{S+C}$) (Fig. S2). Predicted values of TOC and $OC_{S+C}$ smaller than 0 were set to 0. The $OC_{S+C}$ to TOC ratio was calculated for all measured and modeled data. Some of the modeled data had an unrealistically low $OC_{S+C}$ to TOC ratio (Fig. S3). Therefore, we removed all samples with an $OC_{S+C}$ to TOC ratio below 0.5 (n = 7). The resulting dataset was only used in the statistical analysis to determine the impact of land use on $OC_{S+C}$ along the soil profile (Section 2.6.1).

## 2.4 Statistical analyses

All statistical analyses were performed using R (R Core Team, 2022).

### 2.4.1 Effect of land use on the concentration of stabilized organic carbon along the soil profile

The depth profiles of $OC_{S+C}$ were compared for soils under forest and agricultural land use. The dataset was tested for outliers using the boxplot method with the R function "identify_outliers" and one sample was removed from the analysis (Kassambara,
2022). In order to get a normal distribution, the $OC_{S+C}$ data were transformed using box-cox transformation with λ = 0.4 using the R function "bcPower" (Fig. S4) (Fox and Weisberg, 2019). Homogeneity of variance was assessed and no evident relationship between residuals and fitted values was observed (Fig. S5). The difference in $OC_{S+C}$ between both land uses along the depth profile was assessed by performing multiple analyses of variance (ANOVA) over 20 different depth intervals. As the number of data points decreased with depth, the depth layers were chosen in such a way that they all contained 40 ± 5
samples (i.e., 20 ± 5 forest samples and 20 ± 5 agriculture samples). Consequently, the depth intervals for which the ANOVAs were applied had different thicknesses (between 10 cm in the topsoil and 160 cm for the deepest layer) and overlap with each





other (Fig. 2; Table S1). For each depth layer, an ANOVA was performed. If the confidence interval contained 0, the difference between forest and agriculture was considered not to be significant (Table S1).

### 2.4.2  Soil characteristics controlling the concentration of stabilized organic carbon

A statistical model was created to determine the most important variables controlling the concentration of stabilized OC along the sampled depth profiles under forest and agriculture. For this purpose, a generalized additive mixed effects model (GAMM) was constructed. The model included $OC_{S+C}$ as the response variable and selected characteristics of the samples as predictor variables (i.e., land use, texture, weathering indices, local hillslope gradient, $Al^{3+}$ content, and pH), and a smoother for depth and a power variance function on depth (more detail in Supplementary information 1; Fig. S5; Table S2). To construct the

GAMM, only the 116 samples on which soil properties were analysed in the lab were used, as MIR-projection was not available for all soil characteristics.

The predictor variables in the GAMM were tested for correlation and multicollinearity. As the two weathering indicators, the total reserve in bases (TRB) and the chemical index of alteration (CIA), were correlated (Pearson correlation = 0.72), only the

CIA was kept to fit the model (Fig. S7). Because the variance inflation factor (VIF) of Al and CEC was above 5, CEC was removed from the analysis (Fig. S8). An initial GAMM, including all selected predictor variables and realistic interactions between these variables, was fitted using the *gamm* function from the package *mgcv* (Wood, 2011). Using the summary statistics of the LMM part of the GAMM, the variable with the lowest *P*-value was removed and the model was fitted again. This stepwise removal was repeated until the *P*-values of all the remaining variables were significant (i.e., *P*-value < 0.05).

The model was fitted again with standardized variables to allow the comparison among the regression coefficients.

Before validation, the model was tested for the following assumptions: the linear relationship between explanatory and response variables, homogeneity of the variance, and normal distribution of the residuals (Fig. S10). All of these assumptions were confirmed. The final GAMM was validated by comparing fitted values (i.e., the fitted value of the linear mixed effect

(lme) part of the model) with the measured values of $OC_{S+C}$ (Fig.S9). The model performed overall well (i.e., $R^2$ = 0.78, RMSE = 0.29 (% $OC_{S+C}$)), but tended to underestimate the higher values (> 2.5%) of $OC_{S+C}$. The final GAMM was also compared to fitted values of a more complex model (i.e., a model with the same structure, but including all initial variables and interactions) (Fig. S9). The values of both models were similar (i.e., $R^2$ = 0.84, RMSE = 0.26 (% $OC_{S+C}$)), which confirms that the final model performed nearly as well as the more complex model.

### 285  2.4.3  Degree of OC saturation of soil minerals as a function of the silt and clay content

To determine the influence of the portion of <53 µm soil particles (i.e., the S&C fraction) on the maximum $OC_{S+C}$, the boundary line method was used (Fig. 4) (Feng et al., 2013). For different levels of S&C fraction, the samples with $OC_{S+C}$ below a certain threshold (i.e., the boundary line) can be considered to have an $OC_{S+C}$ saturation deficit, while having a physical potential to





stabilize more OC. For performing the boundary line analysis, the samples were grouped according to the proportion of S&C
fraction in the bulk soil. The first category included the three samples with a proportion of S&C fraction lower than 50%. The
samples with more than 50% S&C were separated into five categories with intervals of 10%, resulting in 6 S&C fraction
categories: 0 - 50%, 50-60%, 60-70%, 70-80%, 80-90%, and 90-100%. The top 20% samples having the highest $OC_{S+C}$ of
each group were selected (except for the first group where all data were selected) and used to fit a linear regression of which
the intercept was forced to 0. The regression line was considered as the upper boundary of $OC_{S+C}$ concentration in the soil
(Feng et al. 2013). This boundary line represents the maximum $OC_{S+C}$ concentration at each level of S&C fraction. The
saturation level of $OC_{s+c}$ ([%]) is defined as the ratio of measured $OC_{s+c}$ concentration to the maximum $OC_{s+c}$ concentration at
the corresponding S&C fraction level.

The saturation level of $OC_{s+c}$ under forest agricultural land use were compared for different depth layers (Fig. 5b, Table S6).
The dataset was tested for outliers with a boxplot method using the R function "identify_outliers" and two samples were
removed from the analysis (Kassambara, 2022). In order to get a normal distribution, the stabilized OC saturation data were
transformed using box-cox transformation with $\lambda = 0.38$ using the R function "bcPower" (Fig. S17) (Fox and Weisberg, 2019).
Homogeneity of variance was assessed and no evident relationship between residuals and fitted values was observed (Fig.
S18).

**3    Results**

**3.1 Soil Characteristics and depth profiles**

The collected data showed no specific pattern with depth for most soil characteristics, except for $Al^{3+}$, which increases with
depth (Fig. S12). The grain size varied between the different profiles and tended to be slightly finer under agriculture. The
variability of soil $pH_{H2O}$ was largest in the topsoil and overall profiles, it was between 4.4 and 6.3.

**3.2 Stabilized soil organic carbon for different land uses**

Our results showed that the difference in the concentration of stabilized OC between forest and agricultural soils was only
significantly different in the top 90 cm of the soil (Fig. 2), indicating that deforestation did not significantly affect deeper soil
$OC_{S+C}$. The $OC_{S+C}$ concentration in the forest topsoil (i.e., 0-30 cm depth) was 2.05 ± 0.64%, and it was 1.27 ± 0.56% in the
arable topsoil. The average TOC concentration of the topsoil under forest was 2.50 ± 0.80% and 1.50 ± 0.56% under arable
land (Fig. 3a, Fig. S11). The difference between the two land uses was statistically significant down to 90 cm depth. Below
this depth, the TOC% under arable land and forest was 0.39 ± 0.25% and was, as for $OC_{S+C}$, not statistically significantly
different between the two land uses (Table S4, Table S5).





The impact of land use on $OC_{S+C}$ concentrations below 90 cm could only be tested for soils that were sampled deeper than 90

cm. Therefore, it was necessary to confirm that the above-mentioned result was due to the depth effect, as it could be an artifact

of site subsetting (i.e., if the sites with deeper soils do not have a significant land use effect in their topsoil). To test this, an

ANOVA was performed on the land use effect on $OC_{S+C}$ concentration using a subset (36 sites) containing only the datapoint

of above 90 cm and excluding the profiles that did not have any measurement below 90 cm. The resulting $P$-Value was

significant (i.e., $< 0.05$), reflecting that there is no reason to suspect such a bias.


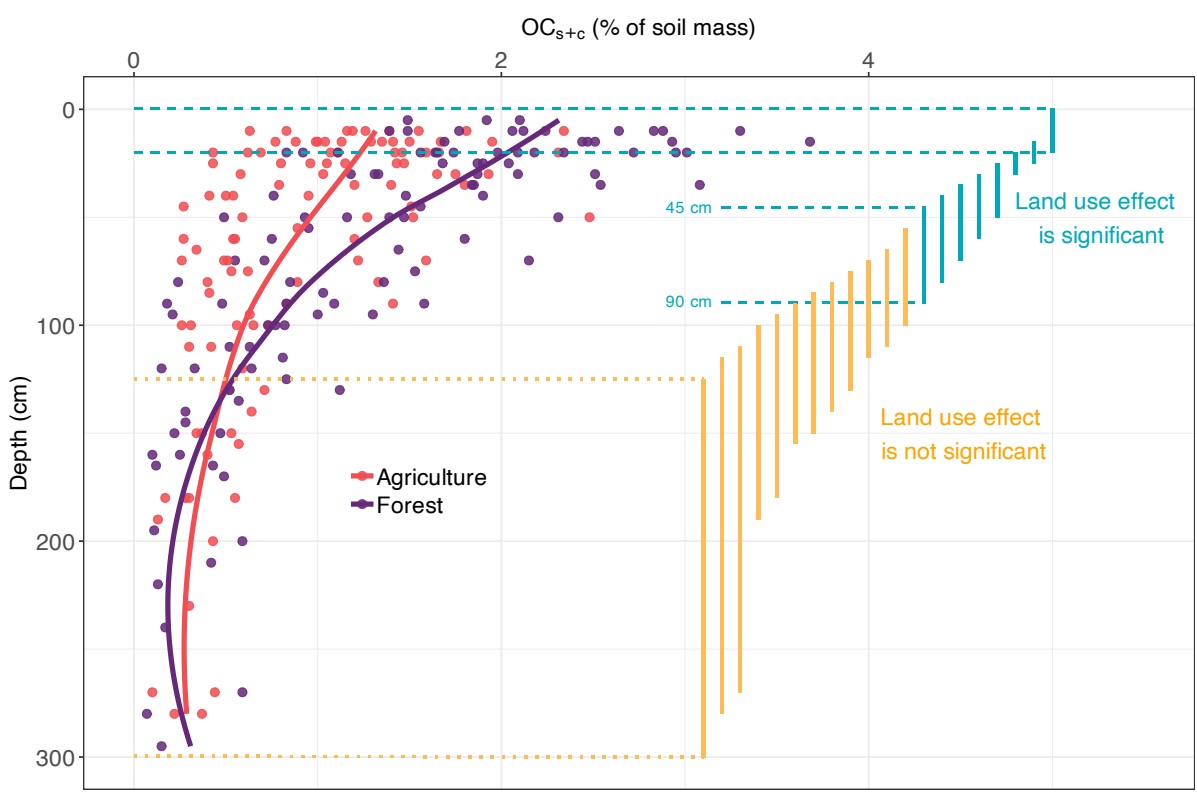

*Figure 2: Silt and clay organic carbon ($OC_{S+C}$) concentration along the depth profile for agricultural and forest soils (dots),*
*with trends indicated by a loess smoother (lines). Vertical bars on the right represent the soil depth layers over which the*
*$OC_{S+C}$ concentrations of both land uses were compared. Blue vertical bars indicate that the difference between the land uses*
*is significant over the considered depth layer, while yellow bars indicate no significant difference.*



(a)

(b)

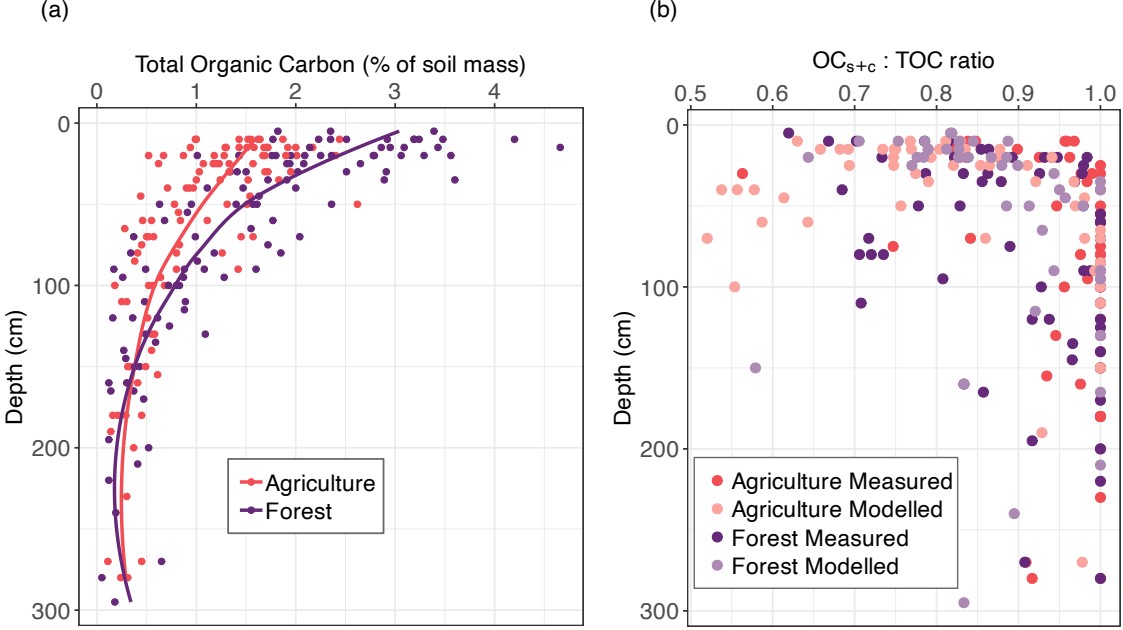

*Figure 3: a) Total organic carbon (TOC) concentration along the depth profile for agricultural and forest soils (dots), with*
*trends indicated by a loess smoother (lines). b) Ratio of the concentration of silt and clay organic carbon (OC$_{S+C}$) over the*
*concentration of TOC along the depth profile. Measured data have been obtained by direct analysis, while modelled data*
*have been predicted using mid-infrared spectroscopy.*

### 3.3 Controlling Factors of the Amount of Stabilized Soil Organic Carbon

The parameters selected for the final GAMM model were depth, land use, clay, silt, $Al^{3+}$, the interaction of land use and pH,
and the interaction of silt and $Al^{3+}$ (Table 1). These parameters are therefore the important controlling factors of $OC_{S+C}$
concentration in these soils. The other parameters (i.e., CIA, pH, and slope) and most interactions were not significant to
describe the $OC_{S+C}$ concentration. The lme part of the model had an $R^2$ of 0.78 and an RMSE of 0.29 (% $OC_{S+C}$) and tended to
underestimate values with higher $OC_{S+C}$ concentration (Fig. S9). The contribution of the smoother to the fitted value is very
high as it ranges between ca. -1 and +1 (% $OC_{S+C}$) and decreases with depth; it is positive above 66 cm and becomes negative
below 66 cm (Fig. S13). This means that the depth smoother increases the slope of the regression curve for depths below 66
cm and decreases the slope of the regression curve above this depth threshold. Therefore, the parameters that influenced the
most the $OC_{S+C}$ concentration were depth and land use.






**Table 1:**

**Statistics of independent variables selected by the GAMM to predict OC$_{S+C}$ concentrations.**

|  | Estimate | Std. Error | t-Value | *P*-Value |
|---|---|---|---|---|
| Intercept | 0.42 | 0.79 | 0.54 | 0.59 |
| Landuse(Forest) | 4.08 | 1.34 | 3.03 | <0.05* |
| pH | -0.83 | 0.61 | -1.36 | 0.18 |
| Clay | 0.39 | 0.11 | 3.57 | <0.05* |
| Silt | 1.07 | 0.37 | 2.88 | <0.05* |
| Al$^{3+}$ | 0.72 | 0.32 | 2.30 | <0.05* |
| Landuse(Forest) :pH | -4.06 | 1.36 | -2.99 | <0.05* |
| Silt : Al$^{3+}$ | -0.72 | 0.29 | -2.51 | <0.05* |

### 3.4 Saturation of Stabilized OC as a function of clay and silt content

The saturation of stabilized OC on soil minerals was higher in the topsoil and the degree of stabilized OC saturation decreased with soil depth under both land uses (Fig. 5a). The saturation was significantly different between the land uses above 50 cm depth, but not below this depth (Fig. 5b, Table S6). The silt and clay content was selected by the GAMM as good predictors for OC$_{S+C}$ concentration and the S&C fraction (i.e., soil fraction < 53 μm) was well adapted for the boundary method as there seems to be a linear correlation between the S&C fraction and the OC$_{S+C}$ for the selected points (Fig. 4). Therefore, the S&C

fraction was used as an independent variable to create a linear model that predicts the maximum OC$_{S+C}$ concentration. The statistical model showed a boundary line (y= 2.8x, with y describing the maximum amount of OC$_{S+C}$ and x the S&C fraction) that enabled to estimate the saturation of stabilized OC for each soil sample (Fig. 5a).






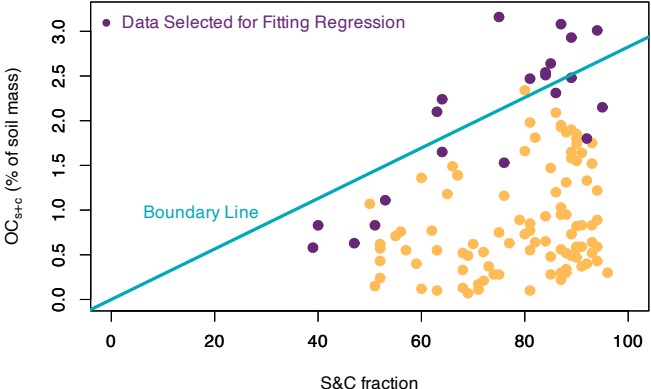

*Figure 4: The silt and clay organic carbon ($OC_{s+c}$) concentration in function of the fraction of the soil particles < 53 μm (S&C fraction). The boundary line (blue) indicates the estimated maximum amount of $OC_{s+c}$ that can be stabilized at any given portion of soil particles in the S&C fraction. The purple dots were used to construct the linear regression for the boundary line.*


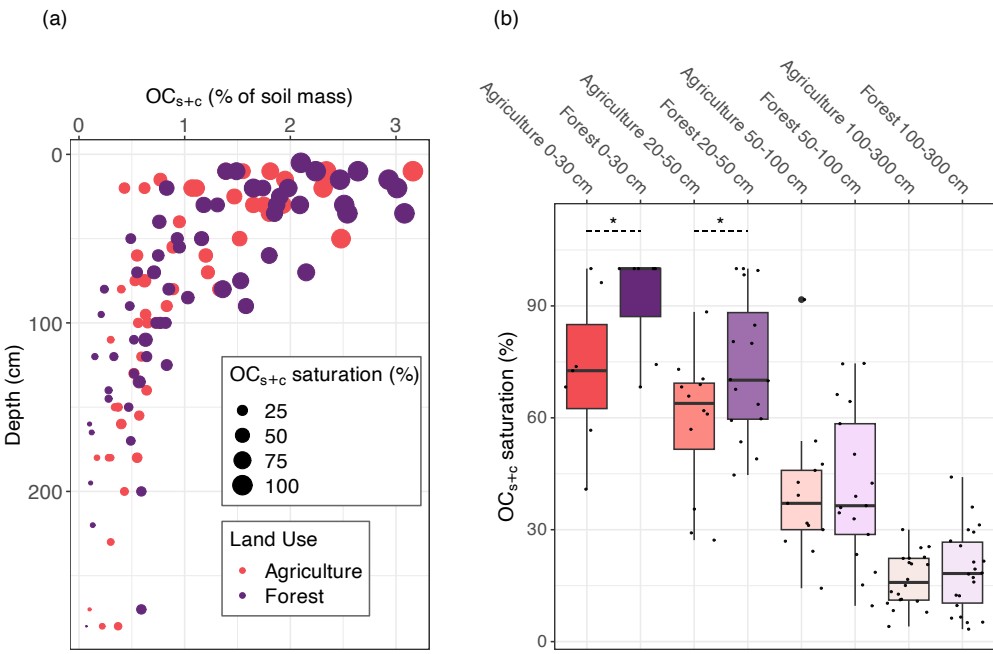

*Figure 5: a) Soil profiles of the saturation level of stabilized OC (i.e., $OC_{S+C}$), expressed as the percentage of the maximum of stabilized $OC_{s+c}$ and indicated by the size of the dots for soils under forest and agriculture. b) Boxplots of the saturation level of $OC_{s+c}$ per depth layer and land use. The asterisks above the plots indicate statistically significant differences between both land uses.*




## 4    Discussion

### 4.1 The Effect of Land use on stabilized organic carbon along the soil profile

Our results show that deforestation during at least 30 years on the studied subtropical soils led to a significant decrease in
stabilized OC in the top 90 cm, while no significant effect on stabilized OC deeper in the soil was detected. Most studies on
the effect of deforestation on SOC stocks focus on the top 10 to 30 cm of the soil, and concern mainly total organic carbon
(TOC). On average, observed relative losses of TOC in tropical topsoils after deforestation range between 10% and 58% (de
Blécourt et al., 2019; Detwiler, 1986; Don et al., 2011; Gurmessa et al., 2016; Kassa et al., 2017; Powers et al., 2011; Rittl et
al., 2017; Veldkamp et al., 2020). According to our results, deforestation led to a decrease in the $OC_{s+c}$ concentration of 45%
in the top 20 cm, 42% between 20 cm and 30 cm depth, and from 27% to 38% in depth layers between 30 cm and 90 cm (Table
S4). The decrease in TOC concentration was very similar to the decrease in the $OC_{s+c}$ concentration (Table S5).

These results indicate that stabilized OC in the studied subsoils is much less dynamic than in the topsoil. This implies that
increasing the amount of OC stored below 90 cm depth in the studied agricultural soils would be a very slow process. This is
evident from the lack of significant differences in $OC_{s+c}$ between both land uses below this depth, showing that the $OC_{s+c}$
concentration at these depths has a slow turnover rate. Hence, reforesting arable land is not likely to lead to an increase in
stabilized SOC below this depth over time scales relative to ongoing climate change, limiting the potential atmospheric $CO_2$
withdrawal by reforestation into the subsoil in this ecosystem. Indeed, we observed that decades after deforestation, the
concentration of OC stabilized in the deeper agricultural soil layers remains largely unchanged, the lowest soil layer with a
significant difference of stabilized OC between forest and crop plantation was located at 45-90 cm depth. Our observation of
a significant effect of deforestation on SOC down to 90 cm depth is a deeper threshold compared to similar studies in the
tropics, in which the difference in SOC content between these land uses was found not to be significant below 30 cm depth
(de Blécourt et al., 2019; Dechert et al., 2004; Gurmessa et al., 2016; Kirsten et al., 2019; Nagy et al., 2018). Nevertheless, our
findings are consistent with the findings of other studies with a similar design (Kassa et al., 2017; Tesfaye et al., 2016) and
emphasize that SOC changes after land-use conversion need to be monitored at least down to 100 cm depth (Strey et al., 2016).
Overall, the results are consistent with the observations of Balesdent et al. (2018); according to their study, OC inputs in deeper
soil layers are hardly affected by land use change, as OC below 100 cm is very old (>1000 years).

Current knowledge of SOC dynamics allows us to hypothesize which factors shaped the effect of land use change on $OC_{S+C}$
concentrations along the depth profile in the studied soils. Concerning the significant difference in $OC_{S+C}$ concentration in the
top 90 cm of the two land uses, a possible hypothesis could be that the difference in $OC_{S+C}$ was due to soil erosion of the arable
soils. During soil erosion, the topsoil is removed and the new upper layers at the eroded site consist of former subsoil that is
typically less saturated in $OC_{S+C}$ compared to topsoil (Doetterl et al., 2015b). However, soil erosion rates in our study area are
low (0.9-1.4 ton $km^{-2}$ $yr^{-1}$) and are therefore unlikely to substantially affect our observations on the considered timescale of a





century (Brosens et al., 2020). A more probable hypothesis is that the different soil $OC_{S+C}$ concentration after deforestation is the result of a change in the quantity and quality of OC inputs. For instance, trees have a much higher root density than crops. Therefore, higher rates of root exudation and root decomposition in forests lead to higher OC inputs to the soil than in arable land (Jackson et al., 1996; Sokol et al., 2019). Furthermore, belowground carbon inputs are more likely to form stabilized SOC than aboveground inputs (Jackson et al., 2017; Sokol et al., 2019; Sokol and Bradford, 2019). This is especially noticeable in

tropical soils, where POM accumulation from litter deposition is limited by the quick decomposition of the organic matter (Sokol et al., 2022). Therefore, the difference in $OC_{S+C}$ concentration over the 0-90 cm depth layer is likely due to the higher below-ground OC inputs in forest soils. Below 90 cm depth, $OC_{S+C}$ is likely to originate primarily from root-derived dissolved OC that slowly leaches downwards along the soil profile while experiencing cycles of sorption and desorption (Uselman et al., 2007; Kaiser and Kalbitz, 2012).

**4.2 Factors controlling the concentration of stabilized organic carbon along the soil profile**

The general additive mixed model (GAMM) showed that information on land use and soil characteristics that are generally used as indicators of $OC_{S+C}$ in tropical regions, such as texture and $Al^{3+}$, can also be used to predict the concentration of $OC_{S+C}$ at soil depths below one meter. However, our results showed no indication of an important control of the weathering degree of soil minerals on the $OC_{S+C}$ concentration. This contrasts with Doetterl et al. (2018), who identified long-term weathering as

a dominant controlling factor for OC storage in soils. Nevertheless, their results were obtained by studying geochemically different soils across a spatial gradient, while the soil samples collected for our work were highly weathered along the soil profile (the interquartile of CIA ranges from 84 to 96) and geochemically similar. This suggests that, while the weathering degree of soils is an important control on potential SOC storage across spatial scales (Doetterl et al., 2018), it does not provide insights into the SOC storage potential along the depth profile of highly-weathered (tropical) soils, highlighting the scale-

specificity of SOC stabilization indicators (Manning et al., 2015).

The obtained controlling factors of the $OC_{S+C}$ concentration are consistent with a large number of studies that emphasize the role of fine soil texture and cations to stabilize OC in soils (Hassink, 1997; Six et al., 2002c; Kunhi Mouvenchery et al., 2012; von Fromm et al., 2021; Zinn et al., 2007). The relative importance of these factors depends on clay mineralogy (Quesada et

al., 2020). Mineralogy, and more broadly soil type, are frequently pointed out as important controlling factors of SOC (Powers et al., 2011; Singh et al., 2018; Wiesmeier et al., 2019). In the literature, climatic variables, in contrast to soil properties, are frequently highlighted as the most important controlling factors of the amount of SOC (Don et al., 2011; Haddix et al., 2020; Hombegowda et al., 2016; Marín-Spiotta and Sharma, 2013; von Fromm et al., 2021). Climatic variables were not considered in this study, because samples were all collected in the same region where no major climatic differences between the sampling

locations are expected (Fig. 1). Besides, studies focusing on subsoils have observed that climatic variables impacts SOC only in the topsoil and not in the subsoil (van Straaten et al., 2015; Luo et al., 2019). Based on our results, we suggest that variables





related to climate and soil weathering are important at the large spatial scales, while at regional and local scales with (similar climate and bedrock characteristics), soil texture becomes more important in controlling the amount of stabilized SOC.

### 4.3 Maximum amount of stabilized organic carbon as a function of clay and silt content

To estimate the potential of the studied soils to stabilize OC, the S&C fraction (i.e., the soil fraction < 53 µm) of samples was used as an independent variable to estimate the upper limit of $OC_{S+C}$ concentration using the boundary line method (Feng et al., 2013). The slope of the boundary line is 2.8 %$OC_{S+C}$ per unit silt and clay fraction (Fig. 4). This is a gentle slope when compared to the result of the global analysis including multiple land uses from Georgiou et al. (2022), who used a similar approach. These authors found a slope between 4.3 and 5.1 for low-activity minerals (i.e., kaolinite, gibbsite), which are

dominant at our study site (Ito and Wagai, 2017). This suggests that the increase in $OC_{S+C}$ concentration with S&C fraction is lower for the studied soils compared to the average soil with a similar mineralogy. However, also land use plays a role in the magnitude of this slope. For example, Georgiou et al. (2022) considered multiple land use types in their study (e.g. grassland, Savana, Schrubland) while Six et al., (2002c) showed that the increase in concentration of OC associated with soil particles of a size < 50 µm was faster for grassland soils than for agricultural or forest soils.


The S&C fraction was selected among other parameters because both silt and clay contents were important controlling factors for $OC_{S+C}$ according to the GAMM. Doing the same analysis using $Al^{3+}$ (which was also an important controlling factor of $OC_{S+C}$ concentration) did not result in a similar relation (Fig. S14). This is due to the increasing concentration of $Al^{3+}$ with decreasing depth, which is common in acidic soils (Kalbitz and Kaiser, 2008). Therefore, although $Al^{3+}$ is a positive controlling

factor of $OC_{S+C}$ concentration, it correlates negatively with it because of the opposite pattern with soil depth (Fig. S12). Using the boundary line method with the chemical index of alteration (CIA) as the independent variable confirmed that also this parameter was not a good indicator of the potential for soils to stabilize carbon for the studied soils. Concerning the relationship between CIA and $OC_{S+C}$ concentration, it would be expected that the higher the CIA, the higher the potential of the soil to store OC. However, the relationship between the CIA having the highest $OC_{S+C}$ concentration and $OC_{S+C}$ concentration is

negative (Fig. S15). Therefore, using the boundary line method using the CIA did not work for the soil samples in our study. This is in line with the result of the GAMM that the CIA does not control the amount of $OC_{S+C}$ along soil profiles.

Our results show that only a small portion of the studied soils had a topsoil $OC_{S+C}$ concentration close to saturation, while most topsoil and all subsoil samples were substantially undersaturated in OC (Fig. 5a, Fig. 5b, Fig. S16). This indicates that deeper

soil layers do not reach their OC storage potential, even under natural vegetation. Therefore, we argue that while there might be a potential to increase the concentration of stabilized OC in the top 90 cm of the studied agricultural soils by converting them back to forest, it does not seem possible to increase the $OC_{S+C}$ concentration in the deeper soil layers over a time scale of decades. This supports multiple studies suggesting that adapted agricultural management techniques and strategic crop breeding have the potential to increase the SOC concentration only in the topsoil (Angers et al., 2022; Chenu et al., 2019; Kell,

2012; Hombegowda et al., 2016; Lynch and Wojciechowski, 2015; Poffenbarger et al., 2020; Sayer et al., 2019). Our data, together with other studies, show that the potential to use the subsoil as an atmospheric carbon sink might be limited to the top 100 cm of the soil (Kirschbaum et al., 2021; Lorenz and Lal, 2005; Mathieu et al., 2015; Rumpel and Kögel-Knabner, 2011). In fact, below 50 cm and 100 cm depth, none of the forest soils reached a degree of $OC_{S+C}$ saturation above 75% and 50% respectively (Fig. 5b, Fig. S16). This indicates that these soil layers are limited by C inputs with respect to $OC_{s+c}$ saturation

and not by the potential of minerals to associate with more OC. Therefore, we argue that using the maximum OC storage potential under natural vegetation as the maximum attainable OC storage below 50 cm depth would not be an appropriate measure to assess potential increases in OC concentration after the conversion of arable land to natural vegetation.

## 5   Conclusion

This study shows that the conversion of forest to agriculture in a subtropical region affected the concentration of stabilized OC

(i.e., $OC_{S+C}$) down to 90 cm depth, while no significant different in the $OC_{s+c}$ between 90 cm and 300 cm were detected. We found a difference in $OC_{S+C}$ concentration of respectively 44.2%, 41.2%, and 27.7% at the depth layers 0-20 cm, 25-50 cm, and 45-90 cm, over a time of at least 30 years. The most important factors controlling the concentration of stabilized OC along the studied soil profiles were land use, texture (i.e., silt and clay content), and aluminium cations ($Al^{3+}$). In these highly weathered soils, the silt and clay content was the best predictor of the maximum potential for SOC storage along the soil

profile, while no effect of soil weathering degree was detected. This shows that while soil mineralogy and weathering status may control maximum mineral-associated OC stocks at large spatial scales, this is not the case for the pedon scale in highly weathered sub-tropical soils. Last, it was shown that while soil profiles below 90 cm were highly under-saturated in stabilized OC under both forest and agriculture, it is unlikely that subsoils below this depth have the potential to store additional stabilized OC by reforestation over decadal time scales, as the $OC_{s+c}$ content was not affected below 90 cm by deforestation.

## Code availability


The code can be provided upon request.

## Data availability

The data used in this study are open access and available at https://figshare.com/s/6a1568d35ce35d273207.

## Authors Contribution

Conception and design of the field campaign: LB, JM, and GG.
Design of the experiment: MVdB and JS.



Performing the experiments: CRM and LB.

Analysis and interpretation of the data: CRM, MVdB, PB.

Writing the manuscript: CRM and MVdB with inputs from all co-authors.

## Competing interests

The contact author has declared that none of the authors has any competing interests.

## Acknowledgements

We thank Britta Jahn-Humphrey for help with soil processing, and Reto Zihlman for statistical advice.

## Financial support

This work was supported by the Swiss National Science Foundation SNSF (Ambizione project number PZ00P2_193617 granted to MVdB).

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
