# Peer review of "The limited effect of deforestation on stabilized subsoil organic carbon in a subtropical catchment"

_EGUsphere, 2023_

## Author Comment (AC1)

Dear Referees,

Thank you very much for your valuable feedback, suggestions and inputs. They will undoubtedly contribute to the enhancement of manuscript's quality.
In this document, you will find the answer to your suggestions highlighted in green, placed directly under your comments.

**Referee #1 (Anonymous)**

In this manuscript, authors evaluated the different of stabilized soil organic carbon (SOC) between forest and agricultural field along the profile down to 3 m in a subtropical catchment. Authors found that stabilized SOC content was not affected by land use below 90 cm, indicating a limited effect of deforestation on stabilized SOC in deep soil. Therefore, authors suggested that deeper soil layer is unlikely to serve as SOC sink for climate mitigation. Authors also found that stabilized SOC was predominantly controlled by land use, depth, silt and clay, and aluminium ion, while soil weathering degree was not relevant. The results support authors' hypothesis that the difference of stabilized SOC between forest and agricultural field below 100 cm depth. While it should be noted that this conclusion only remains valid for the regions with highly weather soils in subtropical regions. This manuscript is generally well written with clear objectives, solid methodology and insightful discussion which meets the requirement for the publication in SOIL. Therefore, several issues still need to be addressed before publication.

Line 12: Please either use SOC or OC throughout the manuscript since they share the same meaning in this manuscript.

Thank you for pointing out this inconsistency; it will be changed in the final manuscript as requested.

Line 16: Please indicate how many soil profiles (and soil samples) were used in this study.

The missing information will be added in the abstract and certainly appreciated by the readers. Thank you.

Lines 26-27: at the scale of the soil profile? It is not clear.

Indeed, thank you for pointing out. It will be replaced by "at regional and local scales (with similar climate, bedrock characteristics, and weathering history)"

Line 51: soil organic carbon can be replaced by SOC.

Yes, indeed. It will be change in the final manuscript.

Lines 68-69: Please address the recent MEMS 2.0 model that uses measured SOC fractions for modelling.

This paper holds considerable importance in the recent literature on the subject. Therefore, including a citation to it in that section would be a beneficial addition.

Zhang, Y., Lavallee, J.M., Robertson, A.D., Even, R., Ogle, S.M., Paustian, K. and Cotrufo, M.F., 2021. Simulating measurable ecosystem carbon and nitrogen dynamics with the mechanistically defined MEMS 2.0 model. Biogeosciences, 18(10), pp.3147-3171.

Line 75: vegetation(Cotrufo The space is missing here.

This will be adapted in the final manuscript. Thank you.

Line 80: Is it necessary to separate SOC from TOC? If they have the some meaning, then the use of SOC would be enough.

Most studies on SOC do not investigate soil fractions and in this context, SOC is essentially indistinguishable from TOC. However, because we study different soil fractions, SOC becomes a more general term, since there is SOC in each of the different soil fractions.
Therefore, in the manuscript we chose to highlight the difference between the SOC from the Silt and Clay fraction ($OC_{S+C}$) and the SOC from the bulk soil (TOC).

Line 85: silt (soil particles in 2- 53 µm) would be better.

This will be adapted in the final manuscript.

Line 95: (Alcántara 95 et al., 2016) the font for this text is different from others. Please correct it.

This will be adapted in the final manuscript.

Lines 143-144: I expect to have more information about the approach for soil sampling design. And there you should indicate how many soil profiles were collected not just the number of soil samples.

This was also pointed out by Referee #2; the sampling design will be described more extensively in the methods section of the final manuscript and its limitation will be addressed in the discussion.

Lines 158-159: How you get the information of weathering degree before laboratory analysis for choosing the sites for laboratory analysis? More detailed information is needed.

Weathering degrees were measured on the soil samples for a previous study (Brosens et al., 2020). Since we used the same soil samples, this information was readily available for our study. This will be more clearly pointed out in the final manuscript.

Lines 227: Grain size is rarely used, please use particle size instead. Then the use of silt (2-53 µm) and sand (53-2000 µm) would be clearer.

This will be adapted throughout the final manuscript.

Lines 239-240: A big concern here is that MIR technique tends to overestimate the low value while underestimate the high value, even the model performance is high. As a result, the high MAOC under forest soil would be underestimated, which potentially leads a close result to the MAOC under agricultural field. I think authors should carefully address this issue in the discussion.

According to Figure S2 (see below), there is no overestimation of the low values of MAOC. However, higher values might indeed be overestimated, and this will be discussed in the final version of the manuscript.

[Figure]

Figure 2: How you conduct the paired comparison at a given depth interval since all the soil profiles were collected from genetic horizons?

The depth intervals for the statistical analyses were created independently from the genetic horizons that were used for sample collection.
The methodology for designing the depth interval is described in lines 253-257:
*The difference in $OC_{S+C}$ between both land uses along the depth profile was assessed by performing multiple analyses of variance (ANOVA) over 20 different depth intervals. As the number of data points decreased with depth, the depth layers were chosen in such a way that they all contained 40 ± 5 Samples (i.e., 20 ± 5 forest samples and 20 ± 5 agriculture samples). Consequently, the depth intervals for which the ANOVAs were applied had different thicknesses (between 10 cm in the topsoil and 160 cm for the deepest layer) and overlap with each other (Fig. 2; Table S1).*

Figure 4: The unit is needed for S&C fractions in the x axis. Please also provide the linear equation here.

The linear equation will be provided, and the unit of the x axis added. Thank you.

Line 378: The effect of land use.

This will be adapted in the final manuscript.

Lines 395-397: What are the potential reasons for the difference between this study and previous studies?

The potential causes are multiple (e.g., study design, data analysis, or location) depending on the study. More detail on this will be added to the discussion in the final version:
- Agricultural fields were too young (Dechert et al, 2004)
- The number of samples was too probably small (e.g., only 11 old agricultural fields in Blécourt et al. (2019), or 10 samples in total in Kirsten et al. (2019))
- Different soil types, climate and/or vegetation (e.g. Blécourt et al. (2019), Gurmessa et al. (2016))

Lines 486-487: why 0-20, 25-50 and 45-90 cm were selected here, instead of 0-20, 20-50, 50-90 cm?

These depth layers were selected because of the methodology to design the depth interval, which aimed to have a similar amount of data for forest and agricultural land use as explained in lines 253-257. This will be pointed out more clearly in the final version of the manuscript.

Lines 489-490: Maximum potential of SOC is predicted by silt+clay, therefore there is no doubt that silt and clay are important.

Indeed. However, silt+clay was selected to predict maximum potential for SOC, because silt and clay were selected as important factors controlling the concentration of stabilized OC by the GAMM (Table 1) and had an adequate relationship with OC for doing a boundary analysis (as opposed to $Al^{3+}$) (Lines 456-458).

**Referee #2 (Prof. Dr. Edzo Veldkamp)**

This is an interesting study in which the authors assess the stabilized SOC content of soils under forest and agriculture in the humid subtropics of Southern Brazil. The samples they use were sampled down to 300 cm in some cases. The laboratory methods that they use are state of the art. They find that the amount of stabilized OC was mostly controlled by land use and soil depth, in addition silt + clay content and exchangeable Al played a role. They cannot show any land use change effect on stabilized OC below 90 cm depth; but do show that the subsoil has not reached above 50% of the 'stabilized OC saturation point'. They conclude that in their study area deforestation does not affect SOC content below 90 cm deep and that it is unlikely that deeper soil layers can serve as an OC sink in timescales relevant for climate change.

The soil samples used in this study were not collected for the purpose that they were used in the present manuscript. Instead they were collected to study how slope gradient affects soil thickness and chemical weathering. I went back to the study by Brosens et al (2020) and learned a few interesting things about the sites which are also relevant for the present study: the study area was chosen because of it relatively homogeneous lithology so that variation in soil depth and weathering are primarily related to topography. Furthermore, site selection was done in a stratified random way, with the goal to cover a wide distribution of slopes. Land use type or history were not accounted for. Only mid-slope positions were sampled. The saprolite in the study area consisted of loose sandy material.

Because of the sampling design that was conducted in this study (largely random), the samples probably included a substantial amount of spatial variability that would have been less if you had collected the samples specifically to detect land use change effects on SOC. We ran into this problem when we conducted sampling for SOC in a montane tropical landscape (de Blecourt et al, 2017). We concluded from this experience that 'scale-dependent relationships between SOC and its controlling factors demonstrate that studies that aim to investigate the land-use effects on SOC need an appropriate sampling design reflecting the controlling factors of SOC so that land-use effects will not be masked by the variability between and within sampling plots'. Compared to the study by de Blecourt et al. (2017), your sampling design has the advantage that you have a relatively homogeneous lithology, nevertheless, your sampling design did not reflect the potential controlling factors of SOC content. I think it is therefore safe to conclude that spatial variability caused a larger variance than would have been the case had you sampled specifically with the aim to detect changes in land use (e.g. by doing paired sampling).

This also brings me to your conclusion that you cannot show land use change effects below 90 cm depth. I was wondering how much this conclusion is also affected by your sampling design? Not being able to show differences means that the variance is too large to detect differences. But as I mentioned earlier your sampling design included spatial variability that would not have been sampled if you had purely focused on differences in land uses. So how large are the chances that this is the case in your study? In a study that we conducted in Indonesia we ran into similar problems (Allen et al 2016). We analyzed this problem by conducting an analysis of variance components and a power analysis. It turned out that in the Allen et al., (2016) study a substantial part of the variance was caused by variance within the replicate plot, which explained why we were not able to show land use related changes in some soil characteristics. Furthermore, we were able to show the optimum sampling size using a power analysis with a power of 80%. My point is you write it correct: you were not able to show any land use change effect on stabilized OC below 90 cm depth, but maybe this was simply caused by you sampling design in combination with the relatively low number of samples in the deeper part of the soils. You can analyze this by conducting a power analysis, which I encourage you to do.

Thanks a lot for the very detailed and precise feedback and suggestions. Your explanation was very clear and helpful for understanding how to improve this work.

Your intuition was correct; according to the power analysis, the variance within the groups is too high in the subsoil, compared to the variance between the groups, and much more samples would be needed to make sure that a type II error is avoided (for example, for the depth 55-100 cm, 90-155 cm, and 115-280 cm, a total of 140 samples, 1469 samples, and 7562 samples, respectively, would be needed to reach a power of 0.8). This is a limitation that needs to be highlighted in the discussion. Table S1 will be modified in the final version of the manuscript, to present the results of the power of the tests (see below).

A more adapted sampling design (e.g. paired plots, sampling at stable landscape locations) would certainly have had a positive impact on the power of the test. However, as the carbon concentration decreases with depth and leads to very small and similar values of OC concentration for both land uses, the impact of the sampling design's quality on the power of the test will certainly decrease in the deeper layer as the variance within the group keeps on increasing as compared to the variance between the groups. We expect that even with the best possible sampling design, the amount of data needed to obtain a power of at least 0.8 would be very large in the deeper soil layers.

**Table S1:**
**Summary of the ANOVA on OC$_{S+C}$ concentration in forest and agriculture for different depth layers**

| Depth layer start at [cm] (included) | Depth layer finishes at [cm] (not included) | Counts Total | p-Value | Significant | Lower Confidence Limit | Upper Confidence Limit | Counts Forest | Counts Agriculture | Power of the test |
|---|---|---|---|---|---|---|---|---|---|
| 0 | 20 | 40 | 0 | Yes | 0.49 | 0.96 | 19 | 21 | 0.99 |
| 15 | 25 | 42 | 0 | Yes | 0.36 | 0.93 | 20 | 22 | 0.99 |
| 20 | 30 | 35 | <0.05* | Yes | 0.29 | 0.91 | 18 | 17 | 0.98 |
| 25 | 50 | 40 | <0.05* | Yes | 0.35 | 0.94 | 18 | 22 | 0.99 |
| 30 | 60 | 41 | <0.05* | Yes | 0.08 | 0.75 | 21 | 20 | 0.71 |
| 35 | 70 | 38 | <0.05* | Yes | 0.19 | 0.93 | 18 | 20 | 0.87 |
| 40 | 80 | 41 | <0.05* | Yes | 0.16 | 0.84 | 18 | 23 | 0.85 |
| 45 | 90 | 42 | <0.05* | Yes | 0.02 | 0.72 | 19 | 23 | 0.57 |
| 55 | 100 | 40 | 0.15 | No | -0.1 | 0.6 | 20 | 20 | 0.32 |
| 65 | 110 | 40 | 0.13 | No | -0.08 | 0.59 | 20 | 20 | 0.34 |
| 70 | 115 | 42 | 0.18 | No | -0.1 | 0.53 | 21 | 21 | 0.28 |
| 75 | 130 | 40 | 0.38 | No | -0.17 | 0.45 | 23 | 17 | 0.15 |
| 80 | 140 | 42 | 0.49 | No | -0.19 | 0.39 | 25 | 17 | 0.11 |
| 85 | 150 | 39 | 0.54 | No | -0.21 | 0.39 | 24 | 15 | 0.10 |
| 90 | 155 | 42 | 0.65 | No | -0.22 | 0.34 | 25 | 17 | 0.08 |
| 95 | 180 | 42 | 0.97 | No | -0.26 | 0.25 | 25 | 17 | 0.05 |
| 100 | 190 | 42 | 0.84 | No | -0.21 | 0.26 | 22 | 20 | 0.05 |
| 110 | 270 | 42 | 0.73 | No | -0.29 | 0.2 | 24 | 18 | 0.06 |
| 115 | 280 | 41 | 0.84 | No | -0.29 | 0.23 | 23 | 18 | 0.05 |
| 125 | 300 | 40 | 0.57 | No | -0.34 | 0.19 | 21 | 19 | 0.09 |

A potential additional problem is that the deeper soil samples that you compared comprised a thicker depth interval. I understand the reasoning to do this: you wanted to have a balanced design when you analyze deeper intervals in the soil. However, since you did not sample the whole depth interval but only discrete samples at different depths, this may have introduced additional variance: Normally SOC contents will decrease with depth and if you consider a depth interval as thick as 160 cm (l. 256) you automatically include systematic variability simply because of the large depth interval takes for your statistical analysis. This again may affect your variance and with that reduce the probability to show differences at larger depths.

Thank you for pointing this out. This was indeed missing in the discussion and will be added in final version of the manuscript.

If I understand it correctly you are comparing depth intervals independent from the depth of the soil. So, if you compare a 30-40 cm depth interval from a shallow soil on a steep slope with the same interval of a deep soil on a flat area, you may actually compare something close to saprolite (shallow soil) with soil material that is extremely weathered (deep soil). Did I understand this correct? If so how much may this have contributed to the variance of your samples? Normally we would like to compare similar soils, you may not have done this if you put shallow and deep soils together.

This is correct, depth intervals were compared independently from the depth of the soil, and this is indeed a limitation that needs to be considered. We have partly addressed this bias by conducting an ANOVA in the top 90 cm of samples with deep soils only (lines 319-324), which showed a significant difference between the two land uses. Also, according to Vanacker et al (2019,see below), we do not observe a strong/abrupt change in soil weathering characteristics close to the saprolite on samples that were collected in the same catchment (see Fig. 4 - upper and middle slope - of their article). This limitation will be addressed in the discussion of the final version of the manuscript.

Vanacker, V., Ameijeiras-Mariño, Y., Schoonejans, J., Cornélis, J. T., Minella, J. P., Lamouline, F., ... & Opfergelt, S. (2019). Land use impacts on soil erosion and rejuvenation in Southern Brazil. *Catena*, *178*, 256-266.

A few minor remarks:

-you mention that you land uses are all >30 years old. According to Table s3 you have also more specific ages, and I would have been very interested to see if your conclusion were the same if you only include sites that are 50 years or older?

Thanks for the interesting question. We did the same statistical analysis as in the manuscript (analysis of variance on land use on different depth layer along the soil profile) but using only the agricultural site that were 50 years or older. According to our results, it would not affect the conclusion, difference between land uses is only observed in the top 90 cm (see table below).

Extra Table: Summary of the ANOVA on $OC_{S+C}$ concentration in forest and agriculture (**that were deforested more than 50 years before samples collection only**) for different depth layers with soil.

|   | LayerLim1 | LayerLim2 | count | pValue | Significant | Forest | Agriculture |
|---|-----------|-----------|-------|--------|-------------|--------|-------------|
| 1 | 0 | 20 | 31 | 0 | Yes | 19 | 12 |
| 2 | 10 | 20 | 28 | 0 | Yes | 16 | 12 |
| 3 | 15 | 25 | 34 | 0.00027 | Yes | 20 | 14 |
| 4 | 20 | 35 | 37 | 0.00055 | Yes | 24 | 13 |
| 5 | 25 | 60 | 41 | 0.00852 | Yes | 25 | 16 |
| 6 | 30 | 70 | 38 | 0.00363 | Yes | 24 | 14 |

| | | | | | | | |
|---|---|---|---|---|---|---|---|
| 7 | 35 | 85 | 42 | 0.01979 | Yes | 25 | 17 |
| 8 | 40 | 90 | 38 | 0.03276 | Yes | 22 | 16 |
| 9 | 45 | 100 | 42 | 0.37522 | No | 27 | 15 |
| 10 | 50 | 100 | 39 | 0.49325 | No | 26 | 13 |
| 11 | 55 | 120 | 42 | 0.08869 | No | 26 | 16 |
| 12 | 60 | 120 | 40 | 0.0854 | No | 25 | 15 |
| 13 | 65 | 130 | 41 | 0.30548 | No | 27 | 14 |
| 14 | 70 | 130 | 39 | 0.49282 | No | 26 | 13 |
| 15 | 75 | 150 | 39 | 0.51353 | No | 28 | 11 |
| 16 | 80 | 155 | 42 | 0.46875 | No | 29 | 13 |
| 17 | 85 | 165 | 42 | 0.81741 | No | 28 | 14 |
| 18 | 90 | 170 | 42 | 0.82409 | No | 29 | 13 |
| 19 | 95 | 220 | 42 | 0.70246 | No | 28 | 14 |
| 20 | 100 | 270 | 42 | 0.89694 | No | 27 | 15 |
| 21 | 110 | 300 | 38 | 0.80436 | No | 27 | 11 |

-All your samples were taking from mid-slope positions (Brosens et al., 2020). How may this have affected the outcome of your study?

The main process that may have affected this is soil erosion, which is generally most intense at mid-slope positions on linear hillslopes. Part of the sediment (and OC) will be deposited on the bottom of the slope, but this will affect mainly the topsoil. However, according to Brosens et al. (2020), erosion rates were generally low and thus, we expect that this effect was limited. This will be added to the discussion in the final version of the manuscript.

-If you sampling design indeed affects your results as much as I anticipate, I suggest that you include more information of your sampling design in your manuscript. It will help the reader to understand the possibilities and limitations of your analyses.

More detail on the sampling design will be added in the final version. Also, the limitation related to it will be discussed.

In summary, I think this is an interesting study but the limitations that are caused by the sampling design may be substantial and should be more prominently discussed. After that is done I expect that it can be published.

Thank you very much for the effort put into understanding this study and its limitations. The potential limitation due to the sampling design and analysis that you pointed out were not yet addressed and doing this will certainly increase the value of this article for the readers.